# tRNA-Derived Small RNAs: Novel Epigenetic Regulators

**DOI:** 10.3390/cancers12102773

**Published:** 2020-09-27

**Authors:** Joonhyeong Park, Se Hee Ahn, Myung Geun Shin, Hak Kyun Kim, Suhwan Chang

**Affiliations:** 1Department of Life Science, Chung-Ang University, Seoul 06974, Korea; marcosweet98@gmail.com (J.P.); mmmk8569@gmail.com (M.G.S.); 2Department of Biomedical Sciences, University of Ulsan College of Medicine, Asan Medical Center, Seoul 05505, Korea; sayahn92@gmail.com; 3Department of Physiology, University of Ulsan College of Medicine, Asan Medical Center, Seoul 05505, Korea

**Keywords:** transfer RNA, tRNA fragment, tRNA-derived small RNA, tsRNA, tRF, tiRNA, epigenetics, cancer

## Abstract

**Simple Summary:**

Cells must synthesize new proteins to maintain its life and tRNA (transfer RNA) is an essential component of the translation process. tRNA-derived small RNA (tsRNA) is a relatively uncharacterized small RNA, derived from enzymatic cleavage of the tRNAs. Accumulating evidences suggest that tsRNA is an abundant, highly modified, dynamically regulated small-RNA and interacts with other types of RNAs or proteins. Moreover, it is abnormally expressed in multiple human diseases including systemic lupus, neurological disorder, metabolic disorder and cancer, implying its diverse function in the initiation or progression of such diseases. In this review, we summarize the classification of tsRNA and its role focused on the epigenetic regulation. Further, we discuss the limitation of current knowledge about the tsRNA and its potential applications.

**Abstract:**

An epigenetic change is a heritable genetic alteration that does not involve any nucleotide changes. While the methylation of specific DNA regions such as CpG islands or histone modifications, including acetylation or methylation, have been investigated in detail, the role of small RNAs in epigenetic regulation is largely unknown. Among the many types of small RNAs, tRNA-derived small RNAs (tsRNAs) represent a class of noncoding small RNAs with multiple roles in diverse physiological processes, including neovascularization, sperm maturation, immune modulation, and stress response. Regarding these roles, several pioneering studies have revealed that dysregulated tsRNAs are associated with human diseases, such as systemic lupus, neurological disorder, metabolic disorder, and cancer. Moreover, recent findings suggest that tsRNAs regulate the expression of critical genes linked with these diseases by a variety of mechanisms, including epigenetic regulation. In this review, we will describe different classes of tsRNAs based on their biogenesis and will focus on their role in epigenetic regulation.

## 1. Introduction

Progress in next-generation sequencing technologies has led to the rapid identification and study of various types of small RNAs [1]. Among them, tRNA-derived small RNAs (tsRNAs) or tRNA-derived fragments (tRFs) are emerging as important RNA regulators [2,3]. A tsRNA is a short noncoding RNA (14 to 40 nt long) generated non-randomly by the action of nucleases on tRNAs [4,5]. Even though proteins are composed of various permutations and combinations of 20 different amino acids, the number of genes coding for tRNAs that carry these amino acids is over 500 [6,7]. 

Accordingly, the variety of existing tsRNAs seems to exceed every estimation till date. Consequently, tsRNA characterization and annotation guidelines have been developed [8,9]. So far, 232 annotated tsRNAs have been included in the Cancer Genome Atlas (TCGA) and the NCI-60 cell line screen databases [10], but this number will certainly increase. 

tsRNA function is an attractive, rapidly expanding area of study within noncoding RNA biology [11]. As a tRNA is a key component of translation, one of the major functions of tsRNAs is most likely translation; a few seminal reports have supported this idea [12,13]. Val-tRF, a tRF from a halophilic archaeon, has been reported to compete with mRNAs for the ribosome, which results in global attenuation of translation in response to stress [14]. In addition, the regulation of RPS28 by LeuCAG3′tsRNA represents a typical mode of regulation by tsRNAs [15]. However, accumulating evidence indicates that tsRNAs play a key role in several processes other than translation; these include maintenance of mRNA stability [16,17], gene silencing [5,18], reverse transcription [19], and gene regulation [20]. These biochemical functions are expected to affect diverse cellular phenotypes. For example, tsRNAs derived from tRNA(Glu), tRNA(Asp), tRNA(Gly), and tRNA(Tyr) suppress cancer cell growth and metastasis [16]. In contrast, tsRNA-26576 promotes breast cancer cell proliferation while inhibiting apoptosis [21], and tRF-03357 promotes ovarian cancer cell growth and migration [22]. In addition, tRF5-AlaCGC enhances the secretion of IL-8 in response to arsenite, via p65 activation [23].

In addition to these functions, accumulating evidences demonstrates that tsRNA has value as a biomarker [17,24,25]. Because the small RNAs are less susceptible to degradation, the detection of tsRNA in various biofluids, such as serum/plasma, urine, and bile is quite feasible [26]. Indeed, a recent report successfully detected tsRNA from 10 kinds of body fluids and showed clear differences in the expression patterns, compared with miRNA [27]. Moreover, some of tsRNA expressions were shown to be correlated with human diseases, including infection [28], prostate/breast cancer [25,29] and myelodysplastic syndromes (MDS) [30]. Despite of this, the detection of tsRNA by conventional small RNA-seq has limitations due to the heavy modification of tsRNA, which inhibits reverse transcription in the RNA-seq procedure [7,31]. With the improvement of accurate detection methods, the tsRNA is anticipated to be used as a diagnostic or prognostic biomarker for various human diseases. 

With this accumulating information regarding tsRNAs, there are critical points that should be further investigated to fully understand the mechanisms of action of these versatile small RNAs. First, there are only few known nucleases that cleave tRNAs, including angiogenin [23], RNase P [32,33], RNase Z [4,5], and Dicer [34], which generate a variety of tsRNAs [26]. Bioinformatic analyses suggest that approximately 200 RNA nucleases are capable of cleaving tRNAs (our unpublished data). Hence, a complex network might be involved in regulating tRNA cleavage (and biogenesis), with implications in cellular physiology and human diseases; this network might warrant further investigation. Second, it is believed that many non-RNA molecules interacting with tsRNAs may exist; these need to be identified. Until now, only a few such molecules have been described, including RNA-binding proteins [16,35], FZD3 [36], and Piwil2 [37]. Considering the complexity of tsRNAs and that they may be induced [23,38], there should also be a group of effectors that directly bind tsRNAs. Lastly, the role of tsRNAs in various human diseases needs to be clarified. A recent study profiling tsRNAs in cancer [25,39], suggested the existence of tissue specific signatures. Further studies will reveal the role of these tissue-specific tsRNAs in cancer initiation and progression.

Since the 1980s, abnormalities in DNA methylation patterns have been identified in cancer cells [40]. Epigenetic mechanisms have often been found to be dysregulated in cancer and have become the focus of cancer research [41,42]. A number of enzymes generating or removing epigenetic marks have been identified through post-translational modification (PTM) mapping. Mutations in these enzymes have been commonly found to be related to cancer. Recently, whole genome/exome sequencing on tumor cells has revealed that epigenetic changes could give rise to cancer, suggesting that epigenetic changes might be involved in tumor suppressor gene inhibition or oncogene activation [42,43]. In addition, several evidences support the association between cancer and epigenetic changes. First, many known tumor suppressor genes have been found to be silenced by hypermethylation [44]. Second, epigenetic writers and readers are required for tumor development in murine cancer models [45,46,47]. Third, changes in DNA methylation patterns are essential for cancer cell survival [48]. These findings highlight the critical role of epigenetic regulation in cancer development.

The role of small RNAs in epigenetic modifications has been relatively well demonstrated in the PIWI protein-interacting RNA (piRNA) pathway. piRNAs guide PIWI proteins toward nascent transcripts of a given transposon and generate heterochromatin by either DNA or histone methylation [49,50]. Interestingly, a pioneering study has revealed that Twi12, a PIWI protein from *Tetrahymena*, carries a tRNA fragment [51]. Besides, it has been revealed that tsRNAs behave like piRNAs [37,52]. Hence, it is possible that tsRNAs are involved in epigenetic regulation along with PIWI proteins. In this review, we summarize the current knowledge on the classification and biogenesis of tsRNAs and on the epigenetic mechanisms involving tsRNAs and their potential applications.

## 2. Classification and Biogenesis of tRNA-Derived Small RNAs

Transfer RNAs (tRNAs) are noncoding RNAs (76–93 nt long) essential for mRNA translation. Each tRNA contains an anticodon sequence that recognizes a specific codon triplet sequence on a mRNA and transfers the corresponding amino acid to a growing polypeptide chain [7]. 

tRNA precursors (pre-tRNAs) are transcribed by RNA polymerase III. They contain 5′ leader and 3′ trailer sequences and undergo splicing, processing, and post-transcriptional modifications (PTMs); a CCA sequence is added at the 3′ termini during the tRNA maturation process [53]. Each mature tRNA contains, on average, 13 modified nucleotides, which affect their stability and functions [54]. tRNAs are folded into L-shaped structures containing three loops (i.e., the D-loop, T-loop, and anticodon loop) (Figure 1).

In humans, mutations, modifications, or dysregulations of tRNAs are associated with several disorders, including myoclonic epilepsy and ragged-red fiber disease (MERRF), mitochondrial encephalomyopathy, lactic acidosis, strokelike episodes (MELAS), and a variety of cancers (breast cancer, lung cancer, cervical cancer, prostate cancer, pancreatic cancer, and multiple myeloma) [55,56]. Moreover, it is becoming increasingly clear that tsRNAs (tRNA- or pre-tRNA-derived small RNAs) are also strongly associated with human diseases, such as amyotrophic lateral sclerosis (ALS) and several cancers, including lung, colorectal, prostate, breast, and ovarian cancers [4,16,21,56,57].

tsRNAs are also called tRFs (tRNA-derived fragments) or tiRNAs (stress-induced tsRNAs) [4,58]. As they are generated by a variety of cleavage events, the classification of tsRNAs is dependent on the site at which tRNAs are cleaved.

### 2.1. Type I tsRNAs: Cleavage at the Stem-Loops (D-Loop and T-Loop) of Mature tRNAs

Type I tsRNAs are generated by cleavage at either the D- or T-loops of mature tRNAs (Figure 1). Type I tsRNAs are classified into 5′ tsRNA (tRF-5 or 5′ tRF) and 3′ tsRNA (tRF-3 or 3′ tRF), which are processed from the 5′ and 3′ ends of mature tRNAs, respectively [5]. The length of the 5′ tsRNAs from different origins, such as HEK293 cells, prostate cancer cells, and *Haloferax volcanii*, is quite heterogeneous. For instance, 5′ tsRNAs of various lengths (14–16 nt (tRF-5a), 22–24 nt (tRF-5b), and 28–30 nt (tRF-5c) long) have been identified. Their 5′ termini coincide with the 5′ ends of the precursor tRNAs, whereas their 3′ termini are either around the D- or the anticodon loops of the precursor tRNAs. In contrast, most 3′ tsRNAs are 18 or 22 nt long. Their 5′ termini are in the T-loop, and their 3′ termini correspond to the CCA sequence that is post-transcriptionally added to the 3′ ends of a tRNA during maturation [59,60]. 

The processing enzymes catalyzing the formation of type I tsRNAs have not been clearly determined. The microRNA-processing enzyme Dicer was first identified as the protein responsible for processing the 3′ ends of the Ile tRNA precursor and type I tsRNAs, such as Gln5′ tsRNA, Gly3′ tsRNA (CU1276), and Lys3′ tsRNA (PBSncRNA) [2,61,62,63]. However, high-throughput sequencing analysis demonstrated that most type I tsRNAs are not processed by any known miRNA-processing enzymes, including Dicer [60,64], suggesting that tsRNA biogenesis might be different from microRNA biogenesis.

### 2.2. Type II tsRNAs: Cleavage at the 3′ Trailer Sequence of Pre-tRNAs

RNase P and RNase Z respectively, process the 5′ leader and 3′ trailer sequences of pre-tRNAs in the nucleus during maturation. Released 3′ trailer sequences become 14–48 nt long tsRNAs, called type II tsRNAs. Their 5′ termini begin after the 3′ ends of the tRNA genomic sequence, whereas their 3′ termini are formed by polyuridine tails, RNA pol III termination signals [5,65].

Although type II tsRNAs are thought to be generated in the nucleus, it has been reported that a type II tsRNA, tRF-1001, is generated in the cytoplasm by ELAC2, a tRNA 3′-endonuclease encoded by a prostate cancer susceptibility gene [4,66]. This finding raises the possibility that type II tsRNAs might originate from multiple biogenesis pathways.

### 2.3. tRNA Halves: Cleavage at the Anticodon Loop of Mature tRNAs

tsRNAs, 30–40 nt in length, that are generated by cleavage at the anticodon loop of tRNAs are called tRNA halves as they are almost half the length of their precursor tRNAs [67]. As previously mentioned, they are also referred to as tiRNAs because they are induced in response to several stress conditions [58]. tiRNAs are simply classified into two groups (i.e., 5′ tiRNAs and 3′ tiRNAs). The 5′ termini of 5′tiRNAs correspond to the 5′ ends of the precursor tRNAs, whereas the 3′ termini are within the anticodon loop of the precursor tRNAs. Consequently, the 3′ ends of 3′tiRNAs are matched to the CCA sequence at the 3′ termini of the precursor tRNAs, and their 5′ ends are a part of the anticodon loop [58,68].

Since the first observation of tiRNAs from *Escherichia coli*, the biogenesis of tiRNAs has been reported in various species, including *Tetrahymena thermophila*, some fungi, and several mammalian cells and tissues [16,23,67,69,70,71,72,73]. The major enzyme involved in the generation of tiRNAs is angiogenin, a member of the RNase A superfamily [58,73]. Under normal conditions, angiogenin is localized within the nucleus or is present in its inactive form in association with RNH1, an angiogenin inhibitor. However, under stress conditions, angiogenin is released from the nucleus or dissociated from RNH1 and cleaves tRNAs at their anticodon loop to process tiRNAs [74,75,76,77]. It has also been reported that angiogenin and Dicer are required for processing tRF5-AlaCGC (tRNA halves) under conditions of arsenite treatment [23].

### 2.4. Cleavage at Other Regions of tRNAs or Pre-tRNAs

Other types of inducible tsRNA, known as i-tRF or tRF-2, include the anticodon-containing internal region of their precursor tRNAs; their length is variable [16,78,79]. Finally, there is a group of tsRNAs whose 5′ ends correspond to the 5′ ends of the leader sequences of pre-tRNAs, whereas their 3′ ends correspond to the 5′ exon halves of the anticodon loops [80]. The detailed biogenesis of these two types of tsRNAs has not been described.

Taken together, tsRNAs can be processed or generated by enzymatic cleavage at specific sites; this demonstrates that these RNAs are not simple degradation by-products and suggests that different types of tsRNAs have distinct biogenesis and may have distinct biological roles as well. Importantly, although hundreds of tsRNAs have been identified using current high-throughput sequencing technologies [9,10,81], no consistent naming conventions have been established, which can serve as a major obstacle in tsRNA research.

## 3. Epigenetic Role of tsRNAs as PIWI-Interacting RNAs

Generally, small RNAs (sRNAs) are classified based on transcript origin, processing pathway, or protein partner. Among sRNA categories, piRNAs, mostly 24–32 nt in length, are found [82]. PIWI proteins belong to the Argonaute/PIWI family, and PIWI-piRNA complexes determine the fertility of various animal species, including mouse, zebrafish, *Drosophila*, and *Caenorhabditis elegans,* via epigenetic regulation of genes involved in developmental pathways in germ and stem cells [83,84,85,86]. Functional studies on the PIWI-piRNA complex suggest that noncoding RNA-associated PIWI proteins may play an important role in epigenetic regulation in germ cells as well as in somatic cells [87]. One of the major studies on tsRNAs focused on the epigenetic role of PIWI-tsRNA complexes, based on the length similarity between tsRNAs and piRNAs as well as their interactions with PIWI proteins.

### 3.1. Role of PIWI-tsRNA Complexes in the Regulation of RNA Processing in the Nucleus

Couvillion et al. demonstrated that type I 3′ tsRNAs associate with PIWI proteins in *Tetrahymena* [88]. They analyzed Twi (*Tetrahymena* PIWI)-interacting small RNAs based on high-throughput sequencing and determined that a variety of unpredictable 23–24 nt long sRNAs were associated with a number of Twi family proteins, such as Twi1, Twi2, Twi7, Twi 8, Twi9, Twi10, Twi11, and Twi12. Interestingly, these sRNAs showed differential dependence on the abundance of distinct Twi proteins [88].

After this initial finding, they identified that 3′ tsRNAs (18–22 nt in length) were associated with Twi12. Structurally, such 3′ tsRNAs begin in the T-loop of the tRNA and end at variable positions of the tRNA 3′-CCA tail. Twi12 is located in the nucleus and is essential for cell growth, suggesting that Twi12-interacting tsRNAs might play a role in cell proliferation as well (Figure 2A) [51].

In a follow-up study, Couvillion et al. demonstrated that the Twi12-3′ tsRNA complexes localized and stabilized Xrn2 within the nucleus and stimulated its exonuclease activity for ribosomal RNA processing [89]. In addition, 3′tsRNAs are required for the nuclear import of Twi12. These findings demonstrate that tsRNAs are novel piRNAs and that tsRNA-PIWI complexes play a role in RNA metabolism in the nucleus (Figure 2A) [89].

### 3.2. Role of PIWI-tsRNAs in Histone Modification in Immune Cells

Zhang et al. showed that tsRNAs modulate histone methylation in human monocytes that can differentiate into dendritic cells (DCs) in response to stimulation by several cytokines, including IL-4 [90]. They performed small RNA deep sequencing and found that tRNA halves, such as tRNA(Glu)-derived piRNA [td-piR(Glu)], td-piR(Gly), and td-piR(Pro), were expressed at higher levels in human monocytes than those in differentiated DCs. These td-piRNAs are approximately 29 nt long and contain a 2′-O-methylated 3′-terminus, as is observed in most piRNAs. Their sequence is identical to the 5′ half of the corresponding mature tRNA, suggesting that they are a type of tsRNA. They found that td-piR(Glu) interacted with a protein from the PIWI family, PIWIL4, and recruited H3K9 methyltransferases (SETDB1 and SUV39H1) and heterochromatin protein 1β (HP1β) to the *CD1A* promoter region. As a result, H3K9 methylation was facilitated at this region, leading to the inhibition of *CD1A* transcription. The authors also identified that IL-4 suppressed tRNA(Glu), resulting in td-piR(Glu) expression. These results demonstrate that tsRNAs are IL-4-regulated signal molecules capable of regulating the chromatin states in immune cells [90]; they also suggest that more tsRNAs might be involved in epigenetic regulation in somatic cells (Figure 2B).

### 3.3. tsRNA Interactions with PIWI Proteins in Human Cancer Cells

Interactions between tsRNAs and PIWI proteins have also been observed in chronic lymphocytic leukemia (CLL) [91]. Type II ts-101 and ts-53 were originally identified as microRNAs (miR-4521 and miR-3676, respectively); they bind to Ago1 and Ago2 proteins and regulate gene silencing. Ts-53 inhibits the expression of T-cell leukemia/lymphoma 1 (TCL1) mRNA by binding to its 3′ UTR. In addition, TCL1 is critical for the development of aggressive CLL. Thus, ts-53 loss derepresses *TCL1* expression, thereby resulting in CLL progression [91]. In addition, Pekarsky et al. found that ts-101 and ts-53 interact with PIWIL2 (human Piwi-like protein), but the function of the tsRNAs-PiwiL2 complex is yet to be determined [52].

## 4. PIWI-Independent Epigenetic Roles of tsRNAs

### 4.1. Regulation of Transposons by tsRNAs

Transposons are genetic elements that can “move” from one location in the genome to another [92]. Genome sequencing has revealed that two classes of transposon elements (TEs) constitute a substantial fraction of most eukaryotic genomes [92,93]. Class I transposons (retrotransposons) can be moved by a “copy and paste” mechanism through the action of a reverse transcriptase enzyme that creates a copy of the TE. This copy is then inserted elsewhere in the genome, leaving behind the original copy. In contrast, class II transposons can move from one place to another by a “cut and paste” mechanism through the action of transposases that cut out a TE from its original location and prepare the new site where it will be inserted [92,94,95].

The transposition activity of TEs contributes to the genetic diversity of all organisms and has been shown to be helpful during adaption to stress conditions [96]. However, TEs can cause genetic mutations and result in the development of several diseases, such as cancers (like colon, breast, ovarian, and liver cancers), neurological disorders (like multiple sclerosis, Rett syndrome, and autism spectrum disorders), and blood diseases [92,94,95,97,98]. 

One of the main roles of piRNA-Piwi protein complexes in germ cells is to protect the genome from the invading TEs by post-transcriptional silencing and DNA methylation [97,99,100,101]. However, piRNA levels gradually decrease, while tsRNAs are enriched during sperm maturation and early embryo development in mice [102], suggesting that tsRNAs might replace the role of piRNA in cells or tissues devoid of piRNA.

Recently, Andrea et al. discovered that 3′ tsRNAs (18 and 22 nt in length) containing the 3′-terminal CCA sequence of mature tRNAs could inhibit long terminal repeat (LTR)-retrotransposons (also known as endogenous retroviruses; ERVs) [103]. It is known that Setdb1 mediates histone H3K9 trimethylation and inhibits most of the LTR-retrotransposons, and that Dnmt1 mediates DNA methylation and inhibits most of the non-LTR-retrotransposons [104,105]. It has been shown that the levels of both 18- and 22-nt-long type I 3′ tsRNAs were elevated in *Setdb1*-/- but not in *Dnmt1*-/- mouse embryonic stem cells (mESCs), suggesting that 3′ tsRNAs might play a role in the regulation of LTR-retrotransposons. Indeed, 18-nt 3′ tsRNAs blocked the reverse transcription of ERVs by competing with mature tRNAs for binding to the PBS (primer binding site) in ERVs. In addition, 22 nt 3′ tsRNAs induce post-transcriptional silencing of ERV mRNA. Taken together, both 18 nt and 22 nt long 3′ tsRNAs suppress LTR-retrotransposon activity by different mechanisms involving sequence complementarity with the PBS sequence (Figure 3A) [103].

### 4.2. Regulation of Chromatin Accessibility by tsRNAs

It is widely known that chromatin accessibility is one of the critical factors in the regulation of gene expression. Accessibility is determined by the state of the chromatin in the form of either heterochromatin (solid form) or euchromatin (fluid form). Moreover, the flexibility of the chromatin regulates the expression levels of relevant genes without altering the DNA sequences [106]. Besides, small noncoding RNAs are involved in chromatin accessibility in *C. elegans* and *Drosophila* [107,108,109]. Especially, siRNAs and piRNAs have been identified as regulators of heterochromatin formation in eukaryotes [110].

Recently, Sharma et al. identified a specific tsRNA (tRNA halves), 5′ tRF-Gly-GCC (tRF-GG), as a suppressor of genes associated with the endogenous retroelement MERVL, which is packaged into and repressed by heterochromatin in embryonic stem (ES) cells and preimplantation embryos [111]. The mechanism underlying this regulation was further determined by Boskovic et al. [35], who discovered that the modulation of tRF-GG levels in human and mouse ES cells affected the production of various RNAs, such as snoRNAs, scaRNAs, and snRNAs, whose stability and function are dependent of Cajal bodies (subnuclear organelles). Among these RNAs, the snRNA U7 is required for processing histone 3′ UTRs via base pairing to the histone downstream element (HDE), thereby regulating histone expression. The modulation of histone expression results in downstream effects on the expression of MERVL-associated genes in murine ES cells and preimplantation embryos. Therefore, tRF-GG affects chromatin accessibility in MERVL elements and throughout heterochromatin [35]. In addition, it was determined that tRF-GG directly bound to heterogeneous nuclear ribonucleoproteins F and H (hnRNP F/H), which are required for normal Cajal body biogenesis, suggesting that hnRNP F/H might play a role in tRF-GG-mediated regulation (Figure 3B) [35].

Studies on the role of tsRNA in the regulation of chromatin accessibility are just in their infancy, and more efforts will be needed to understand the detailed mechanism underlying this process.

### 4.3. The Role of tsRNAs in the Regulation of Adipogenesis

Shen et al. suggested that type I 5′ tsRNAs might act as novel epigenetic molecules regulating adipogenesis [112]. To determine the role of tsRNAs in adipose tissues affected by obesity-induced stress, tsRNA sequencing was performed using perirenal fat from high-fat-diet (HFD)- and low-fat-diet (LFD)-fed mice groups. A total of 296 tsRNAs from the HFD and LFD mice groups displayed differential expression. Among them, 170 tsRNAs were upregulated in the HFD group, whereas 126 tsRNAs were upregulated in LFD group. Such a change in the tsRNA profile suggested that tsRNAs might regulate perirenal fat deposition in vivo [112]. 

tRF^GluTTC^ showed aiding in suppression of 3T3-L1 preadipocyte. Moreover, tRF^GluTTC^ downregulates the mRNA expression of Krüppel-like factor (KLF) family members (KLF9, KLF11, KLF12, and KLF13) [112], which are transcriptional regulators play an important roles in adipogenesis during processes such as cell development and differentiation [113,114,115]. The members of the KLF family play an important role in expression among the 296 differentially expressed tsRNAs between the HFD and LFD groups [112]. tRF^GluTTC^ promotes preadipocyte proliferation via increasing the expression of cell cycle regulatory factors, such as Cyclin D1, CDK4 (cyclin-dependent kinase 4), and Cyclin E [112], which are essential for the maintenance of the G1/S phases in mammalian cells [116,117]. tRF^GluTTC^ also reduces fatty acid synthesis-related gene expression, triglyceride content, and lipid accumulation, lent roles in adipogenesis (Figure 3C) [112]. These findings suggest that tsRNAs could serve as novel therapeutic targets during obesity treatment.

### 4.4. Role of tsRNAs in the Regulation of Intergenerational Inheritance

Intergenerational inheritance refers to the transmission of an epigenetic trait from one generation to the next [118]. It may include nongenetic materials, such as proteins and metabolites, and RNA. The role of tsRNAs in intergenerational inheritance was first identified in a study on HFD-fed mice, in which it was shown that sperm tsRNAs (tRNA halves) contributed to the transmittance of acquired metabolic disorder to the next generation [119]. Likewise, a protein restriction study in mice demonstrated that the upregulation of a 5′ fragment of glycine tRNA resulted in downregulation of the MERVL retroelement both in sperm and egg cells [111]. A more recent report further showed that low-protein diets affected tsRNA biogenesis, thereby supporting the role of tsRNAs in the transmission of metabolic phenotypes [120]. Further, a switch in the RNA payload from piRNAs to tsRNAs in the sperm maturation process indicates that they may be involved in epigenetic inheritance [121]. Related to this, a study using knockout mice showed that the loss of the nucleic acid modification enzyme DNMT2 prevented the intergenerational inheritance of a HFD-induced metabolic disorder [122]. Moreover, it was shown that the absence of DNMT2 resulted in abolished sperm tsRNA modification and expression changes, thereby indicating the critical role of tsRNAs in epigenetic mechanisms.

### 4.5. Translation Regulation by tsRNAs in Cancer

The role of tsRNAs in translation regulation was first reported for tRNA-derived stress-induced fragments (tiRNAs) that inhibit translation initiation by displacing the eIF4F complex from mRNAs [123]. Likewise, pseudouridylation of type I 5′ tsRNAs by PUS7 was reported to control the function of the translation initiation complex in hematopoietic stem cells [31]. Another study revealed an alternative pathway in which the 5′ tsRNA inhibited translation in a sequence-independent manner [12]. A recent report supported this finding by showing that type I 5′ tsRNA interacts with the human multisynthetase complex (MSC) [124], which is critical for translation elongation. tsRNAs interfere not only in the translation initiation/elongation step, but can also enhance translation efficiency through ribosomal RNA processing [13]. Regarding this regulation mechanism, the LeuCAG3′ tsRNA (type I) that is upregulated in hepatocellular carcinoma interacts with some ribosomal protein (RPS15 and RPS28) mRNAs to increase their production.

Interestingly, the role of tsRNAs in cancer was initially proposed from a bladder cancer-derived tRNA fragment that inhibited endothelial cell growth [125]. In addition, early reports on tsRNAs suggested that they have a function in cancer [77]. Several studies identified tsRNA signatures in breast cancer [25,78], and pan-cancer studies that followed also detected some signatures [52,78], suggesting that deregulated tsRNA production plays critical roles in cancer. Indeed, an early report on B-cell lymphoma described a tRNA-derived microRNA, namely, CU1276 (type I tsRNA), that represses RPA1 expression, thereby regulating the DNA damage response [63]. Likewise, tRF5-Glu (tRNA halves) was reported to suppress BCAR3 expression by binding to its 3′ UTR, resulting in the inhibition of ovarian cell proliferation [126]. A recent study also showed that type I 3′ tRF-3019a plays an oncogenic role in gastric cancer, via the regulation of the tumor suppressor FBXO47 [127]. Most of these regulation processes involve microRNA-like repression, mainly via UTR interaction [91]. Considering the critical role of tsRNAs in translation, further investigation on the role of these molecules in various types of cancer will expand our understanding of the oncogenic or tumor-suppressive functions of tsRNAs.

## 5. Conclusions

The roles of tsRNAs in cancer constitute a rapidly emerging area of study. One of the unique characteristics of tsRNAs is their ability to interact with either Argonaute or PIWI proteins [128]. This implies that the functions of tsRNAs may be more diverse than those discovered so far. As described here, tsRNAs are mainly involved in ribosomal and translation regulation, but their epigenetic modification capacities indicate that they also play important roles in immune cells, germ cells, adipocytes, and cancer cells (summarized in the Table 1). 

Similar to other small RNAs, antisense oligonucleotide technologies have been used to inhibit tsRNAs in multiple studies [13,81]. Despite the existing limitations in the development of these antisense tsRNA inhibitors, future research will continue this effort to control the expression of specific tsRNAs. Furthermore, investigating key factors in the biogenesis of cancer-related tsRNAs will provide alternative approaches to regulate abnormal tsRNA levels. Especially, through the identification of specific nucleases responsible for tRNA cleavage and the development of specific inhibitors, it will be possible to control the abnormal production of tsRNAs related to human diseases. Considering that cancer development has generally been linked to elevated translation rates and consequently, to high levels of tRNAs, understanding how cancer cells manage tsRNA levels and avoid unexpected side effects is also an important issue to pursue. These studies will collectively help us understand the role of tsRNAs in epigenetic modification and will enable us to develop novel cancer treatments.

## Figures and Tables

**Figure 1 cancers-12-02773-f001:**
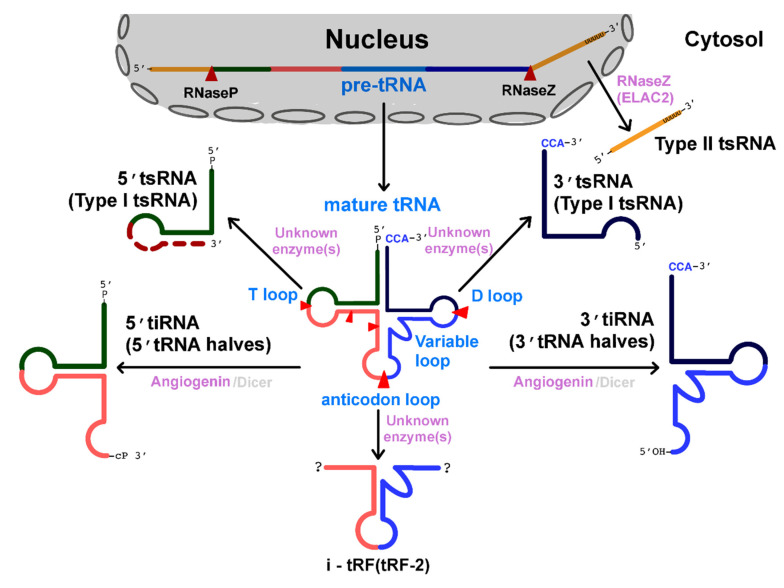
Classification and biogenesis of tRNA-derived small RNAs. More than 6 types of tsRNAs have been classified based on the site at which precursor tRNAs are cleaved. tsRNAs are formed by the processing of their precursors (i.e., mature tRNAs or pre-tRNAs). Cleavage sites are indicated by red arrowheads. Variations in length observed in 5′ tsRNAs (Type I tsRNAs) are indicated by a red dotted line. tRNA halves are primarily processed by angiogenin and Dicer proteins (the name of the minor processing enzyme is shown in gray). The major processing enzymes for other tsRNAs generated from mature tRNAs have not been determined.

**Figure 2 cancers-12-02773-f002:**
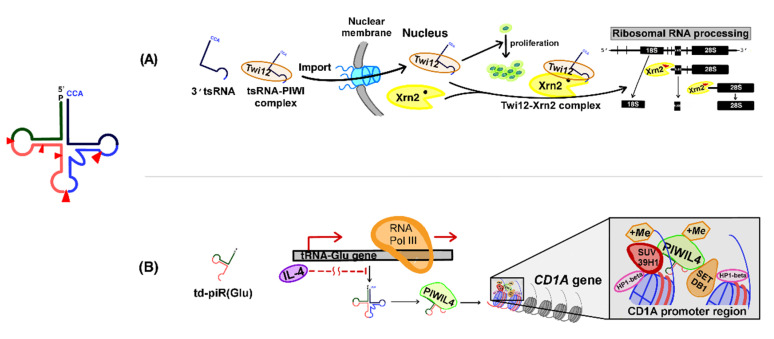
PIWI-dependent epigenetic regulation and function of tRNA-derived small RNAs: (**A**) Regulation of RNA processing in the nucleus. The Twi12-3′ tsRNA complex plays a role in cell proliferation. It also plays a role in the stabilization and nucleolar localization of Xrn2, a 5′ monophosphate-dependent nuclear exonuclease that is required for ribosomal RNA processing in the nucleus. (**B**) Histone modification in immune cells. IL-4 affects tRNA(Glu) expression; consequently, it inhibits the expression of the tRNA(Glu)-derived piRNA (td-piR(Glu)) precursor. td-piR(Glu) associates with PIWIL4 and recruits H3K9 methyltransferases (SETDB1 and SUV39H1) and heterochromatin protein 1β (HP1β) to the CD1A promoter region, thereby facilitating H3K9 methylation. Hence, the transcription of CD1A is inhibited.

**Figure 3 cancers-12-02773-f003:**
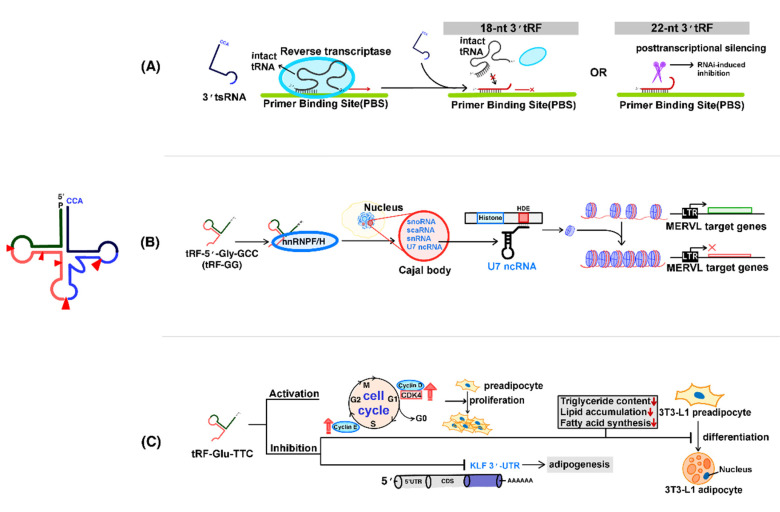
PIWI-independent epigenetic regulation and function of tRNA-derived small RNAs (**A**) Regulation of transposon activity. Intact mature tRNAs play a role as primers during the reverse transcription of ERVs; 3′ tsRNAs (18-nt long; red line) interrupt this process by competing for the primer binding site (PBS), whereas 22 nt long 3′ tsRNAs induce post-transcriptional silencing of retroviral proteins by targeting their mRNA. (**B**) Regulation of chromatin accessibility; 5′tRF-Gly-GCC (tRF-GG) directly binds to heterogeneous nuclear ribonucleoproteins F and H (hnRNP F/H), thereby forming a complex. This complex is required for the biogenesis of a normal Cajal body. Further, U7 snRNA is required for histone expression by base pairing with the histone downstream element (HDE). Consequently, elevated histone levels affect the expression of MERVL-associated genes via altering the chromatin status (euchromatin to heterochromatin). (**C**) Regulation of adipogenesis. Increased levels of tRF^GluTTC^ induce the expression of several cell cycle regulatory factors, such as Cyclin D1, CDK4, and Cyclin E, thereby promoting preadipocyte proliferation, while inhibiting 3T3-L1 preadipocyte differentiation. tRF^GluTTC^–mediated inhibition results in reduced expression of fatty acid synthesis-related genes and a decrease in triglyceride content and lipid accumulation. Finally, increased levels of tRF^GluTTC^ inhibit result in the inhibition of mRNA translation of Krüppel-like factor (KLF) family members through hybridization to KLF 3′ UTR.

**Table 1 cancers-12-02773-t001:** Summarized types and function of tsRNA(s) related to epigenetic regulation.

Type of tsRNA(s)	Common Name	Example of tsRNA(s)	Function	Ref. Number
Type I	5′ tsRNA	tRF^GluTTC^	- Preadipocyte proliferation via increasing the expression of cell cycle regulatory factors - Reduces fatty acid synthesis-related gene expression adipogenesis	[112]
	3′ tsRNA	Twi12-interacting 3′ tsRNAs	- RNA metabolism (stimulated Xrn2 exonuclease activity)	[51]
		18-nt 3′ tsRNA	- Interrupts reverse transcription of ERVs by competing for the primer binding site (PBS)	[103]
		22-nt 3′ tsRNA	- Induces post-transcriptional silencing of retroviral proteins by targeting their mRNA	
		LeuCAG3′ tsRNA	- Translation regulation of ribosomal protein	[13]
Type II		ts-53 & ts-101	- Loss derepresses TCL1 expression - Interacts with PiwiL2	[52,91]
		tRF-1001	- Regulation of cell cycle	[4]
tRNA halves	5′ tiRNA	td-piR(Glu)	- Inhibition of CD1A transcription	[90]
		Sperm 5′ tRNA halve(s)	- Transmission of metabolic phenotypes - Involved in epigenetic inheritance	[119,120,121,122]
		Sperm 5′ tRF-Gly-GCC	- Downregulation of the MERVL retroelement - Affects chromatin accessibility	[111]

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
