# Peer review of "tRNA-Derived Small RNAs: Novel Epigenetic Regulators"

_cancers, 2020, doi:10.3390/cancers12102773_

Round 1

Reviewer 1 Report

The manuscript is focusing on explaining classification and Biogenesis of various tRNA-derived Small RNAs. Also, authors describe epigenetic roles of tsRNAs as PIWI-interacting RNAs and as PIWI-independent RNAs. In overall, the review is quite well written and interesting, but I would like to point out several points for the manuscript.

  1. Please double check the references in the whole manuscript. There are several incorrect citations. For example, in line 37, page 1, it looks like that the reference [9] is not properly cited. The reference should be cited with the reference [8] in line 35, page 1. Please check and correct the reference.
  2. In the figure 1, it would be great that the three loops (D-, T- and anti-codon loop) are annotated in the mature tRNA figure as described in the main text in line 96-97, page 3.
  3. Authors describes more than 6 types of tsRNAs were identified. Although authors showed the different types of tsRNAs in the figure 1, it would be great to show the summary of the different types of tsRNAs in a table which include information about the details about tsRNAs types such as how many tsRNAs in different types / relevant diseases / potential functions / references etc…
  4. There are some typos in the manuscript.
    1. In line 42, page 1: translational => translation
    2. In line 50, page2, tRF5-Ala-CGC => tRF5-AlaCGC
    3. In line 91, page 2: an mRNA => a mRNA

Author Response

Reviewer #1

The manuscript is focusing on explaining classification and Biogenesis of various tRNA-derived Small RNAs. Also, authors describe epigenetic roles of tsRNAs as PIWI-interacting RNAs and as PIWI-independent RNAs. In overall, the review is quite well written and interesting, but I would like to point out several points for the manuscript.

  1. Please double check the references in the whole manuscript. There are several incorrect citations. For example, in line 37, page 1, it looks like that the reference [9] is not properly cited. The reference should be cited with the reference [8] in line 35, page 1. Please check and correct the reference. >> We appreciate for the critical comment. Following the comment, we moved Ref [9] to the sentence before the previous one.
  2. In the figure 1, it would be great that the three loops (D-, T- and anti-codon loop) are annotated in the mature tRNA figure as described in the main text in line 96-97, page 3.>> We agree with the point. Thus, we edited the Figure 1 and annotated D-, T- and anti-codon loops, indicated by bold, blue script.
  3. Authors describes more than 6 types of tsRNAs were identified. Although authors showed the different types of tsRNAs in the figure 1, it would be great to show the summary of the different types of tsRNAs in a table which include information about the details about tsRNAs types such as how many tsRNAs in different types / relevant diseases / potential functions / references etc…>> We thank for the helpful suggestion and agree that such table can concisely deliver important information of tsRNA. Following the comment, we added a summarizing table in the conclusion section.
  4. There are some typos in the manuscript.
    1. In line 42, page 1: translational => translation
    2. In line 50, page2, tRF5-Ala-CGC => tRF5-AlaCGC
    3. In line 91, page 2: an mRNA => a mRNA >> We appreciate for the detailed comments and corrected all typos.

Reviewer 2 Report

This is an overall well-written review paper nicely summarizing the emerging roles of tsRNAs in human diseases and in epigenetic regulation. I have only one suggestion that the authors may add a papragraph summarizing the studiy of tsRNAs in the serum/plasma and their association with diseases, for example (BMC Genomics 2013, PMID: 23638709; J Mol Cell Biol 2014, PMID: 24380870;). Also, tsRNAs has been widely discovered in other human Biofluids (Cell Rep 2018, PMID: 30380423) and could be harnessed as diseases biomarker as well (Biomark Cancer 2018, PMID: 29497340)

Author Response

Reviewer #2

This is an overall well-written review paper nicely summarizing the emerging roles of tsRNAs in human diseases and in epigenetic regulation. I have only one suggestion that the authors may add a paragraph summarizing the study of tsRNAs in the serum/plasma and their association with diseases, for example (BMC Genomics 2013, PMID: 23638709; J Mol Cell Biol 2014, PMID: 24380870;). Also, tsRNAs has been widely discovered in other human Biofluids (Cell Rep 2018, PMID: 30380423) and could be harnessed as diseases biomarker as well (Biomark Cancer 2018, PMID: 29497340)

>> We appreciate for the valuable suggestion and agree that mentioning the potential of tsRNA in biofluids as a biomarker will provide useful information to the field. Thus, we added a paragraph regarding this in the introduction section and included all the suggested references. Thanks for the helpful comment that made this review better.

Reviewer 3 Report

The review "tRNA-derived small RNAs: novel epigenetic regulators" by Park et al., describes the remarkable properties of small RNAs derived by transfer RNAs. The review is interesting and well written, and I suggest its publication with few minor changes.

pag. 1 row 42 the Authors should substitute 'translational' with 'translation'

Fig. 1: left side the Authors should correct 3' tiRNA with 5' tiRNA; in the central scheme it should be useful to indicate the D and T loops; in the lower part it would be more correct to substitute 'unknown enzyme' with 'unknown enzyme(s)'

pag. 3 row 114 the Authors should substitute 'Variations' with 'Variations in length'

pag. 5 row 196 the Authors should add the reference to Figure 2a

Fig. 2: it is really a complicate picture with a lot of details: the Authors should consider dividing it into 2 separate figures.

chapter 3: after the nice scheme reported in Fig. 1 the Authors should consider inserting in every paragraph (in the title or in other parts) the reference to the type of tsRNA they are describing: Type I, Type II, Halves...

Author Response

Reviewer #3

The review "tRNA-derived small RNAs: novel epigenetic regulators" by Park et al., describes the remarkable properties of small RNAs derived by transfer RNAs. The review is interesting and well written, and I suggest its publication with few minor changes.

pag. 1 row 42 the Authors should substitute 'translational' with 'translation'

>> We thank to the helpful comment and corrected it to translation.

Fig. 1: left side the Authors should correct 3' tiRNA with 5' tiRNA; in the central scheme it should be useful to indicate the D and T loops; in the lower part it would be more correct to substitute 'unknown enzyme' with 'unknown enzyme(s)'

>> We agree to the correction and suggestions. We corrected 3' tiRNA with 5' tiRNA and marked D and T loops by blue-bold script. Also, we corrected unknown enzyme to unknown enzyme(s).

pag. 3 row 114 the Authors should substitute 'Variations' with 'Variations in length'

>> We appreciate for the helpful point and corrected it to Variations in length.

pag. 5 row 196 the Authors should add the reference to Figure 2a

>> Thank for the detailed review. Following the comment, we added reference to Figure 2a (Ref #89)

Fig. 2: it is really a complicate picture with a lot of details: the Authors should consider dividing it into 2 separate figures.

>> We agree that the Figure 2 is complex picture with too much information. Following the suggestion, we divided it into two figures (Figure 2 and Figure 3). Thanks for the helpful comment.

chapter 3: after the nice scheme reported in Fig. 1 the Authors should consider inserting in every paragraph (in the title or in other parts) the reference to the type of tsRNA they are describing: Type I, Type II, Halves...

>> We thank for the helpful comment and indicated the type of tsRNA in each paragraph of section 3 and 4.
